# Single-atom tailored atomically-precise nanoclusters for enhanced electrochemical reduction of $CO_2$-to-CO activity

Yi-Man Wang [1], Fang-Qin Yan [1], Qian-You Wang [1], Chen-Xia Du [1], Li-Ya Wang [2], Bo Li [2], Shan Wang [1] ✉ & Shuang-Quan Zang [1] ✉

The development of facile tailoring approach to adjust the intrinsic activity and stability of atomically-precise metal nanoclusters catalysts is of great interest but remians challenging. Herein, the well-defined $Au_8$ nanoclusters modified by single-atom sites are rationally synthesized via a co-eletropolymerization strategy, in which uniformly dispersed metal nanocluster and single-atom co-entrenched on the poly-carbazole matrix. Systematic characterization and theoretical modeling reveal that functionalizing single-atoms enable altering the electronic structures of $Au_8$ clusters, which amplifies their electrocatalytic reduction of $CO_2$ to CO activity by ~18.07 fold compared to isolated $Au_8$ metal clusters. The rearrangements of the electronic structure not only strengthen the adsorption of the key intermediates *COOH, but also establish a favorable reaction pathway for the $CO_2$ reduction reaction. Moreover, this strategy fixing nanoclusters and single-atoms on cross-linked polymer networks efficiently deduce the performance deactivation caused by agglomeration during the catalytic process. This work contribute to explore the intrinsic activity and stability improvement of metal clusters.

Catalysis with well-defined metal nanoclusters (NCs), at the borderline of small-molecule catalysis and heterogeneous metal catalysis, is a rising topic of interest[1–5]. Despite their structural appeal, the generally poor stability and catalytic properties of NCs put them out of reach of practical applications. Extensive research efforts have been devoted to improving the intrinsic catalytic properties of metal NCs by tailoring their compositions and structures, such as heterometallic doping[6–10], ligand engineering[9,11–15], and size regulating[16–18]. While a large number of studies have been released, the controllable synthesis of target metal NCs with well-defined structure still faces great challenges for the synthetic chemists because metal clusters are susceptible to external stimulus during the assembly process[19–21]. Thus, the goal of achieving a metal NCs-based catalyst with high activity and stability raises the necessity for the development of new tailoring approach.

Significant advances have been witnessed in the single-atom (SA) catalysts as efficient catalysts for various reactions[22–26]. Owing to the different electronic states between NCs and SAs[27,28], recent research has found that integrating SAs and metal NCs into one system is an efficient strategy to improve catalytic performance[27–37]. Introducing SAs can result in the asymmetric electron distribution and moderate free energy for intermediates adsorption, thus optimizing the catalytic activity of metal NCs[33,36]. Therefore, modulating the electronic structures of metal NCs through SAs could achieve synergistic effects for a given reaction. However, strategies to prepare nanocluster-single atom (NC-SA) catalysts usually involve pyrolysis treatment[27,29,30,32–35,37]. The resulting catalyst with vague structure makes it difficult to decipher structure–activity relationships. To the best of our knowledge, studies of functionalizing atomically-precise metal NCs with extra SAs to pursue remarkable activity and stability are still in their infancy.

[1]Henan Key Laboratory of Crystalline Molecular Functional Materials, College of Chemistry, Zhengzhou University, Zhengzhou 450001, China. [2]College of Chemistry and Pharmaceutical Engineering, Nanyang Normal University, Nanyang 473061, People's Republic of China. ✉e-mail: shanwang@zzu.edu.cn; zangsqzg@zzu.edu.cn

Herein, uniformly dispersed metal NCs modified with SA sites loaded in a poly-carbazole matrix (denoted as Poly-(Au$_8$-DCP@M), M = Fe, Co, Ni, Cu, Zn) were achieved through an effective co-eletropolymerization strategy (Fig. 1a and Supplementary Fig. 1). The carbazole-substituted phenanthroline composed of chelated metal sites (DCP@Fe, DCP = 3,8-di(9H-carbazol-9-yl)-1,10-phenanthroline) enabled integrating isolated SAs into the hybrid material (Supplementary Fig. 2). Considering that gold-based nanomaterials are intrinsically active for catalyzing CO$_2$ reduction reaction (CO$_2$RR)[16,17,38,39], Au$_8$ NC [Au$_8$(dppp)$_4$(CzPA)$_2$]$^{2+}$ capped by bidentate pisphosphine (dppp = 1,3-bis(diphenylphosphino) propane) and alkynyl carbazole ligands (HCzPA, 9-(4-ethynylphenyl)carbazole) was chosen as the model NC. In comparison with previously reported NC-SA catalysts, the benefit of this strategy is the assembly of monomers will fully pre-disperse the well-defined metal NCs and SAs, thus making them evenly distributed in the resulting skeleton, which providing ideal platforms to understand the critical roles of NCs and SAs in catalytic processes. Remarkably, the as-prepared Poly-(Au$_8$-DCP@Fe) catalysts exhibited much better catalytic efficiency than the pristine Au$_8$ crystal, self-polymerized Poly-Au$_8$, Poly-DCP@Fe and Poly-(Au$_8$-DCP) (co-polymerized Au$_8$ and DCP) toward electrochemical CO$_2$RR. By combining the spectroscopy analysis and theoretical calculations, the introduction of isolated Fe SAs is capable of regulating the electronic structure of Au$_8$ NCs, thus optimizing the adsorption of COOH* in the rate-determining step of CO$_2$RR and accelerating the kinetic processes. This facile strategy of introduction extra SAs to manipulate the intrinsic activity of metal NCs provides valuable insights for designing highly active metal NCs-based catalysts.

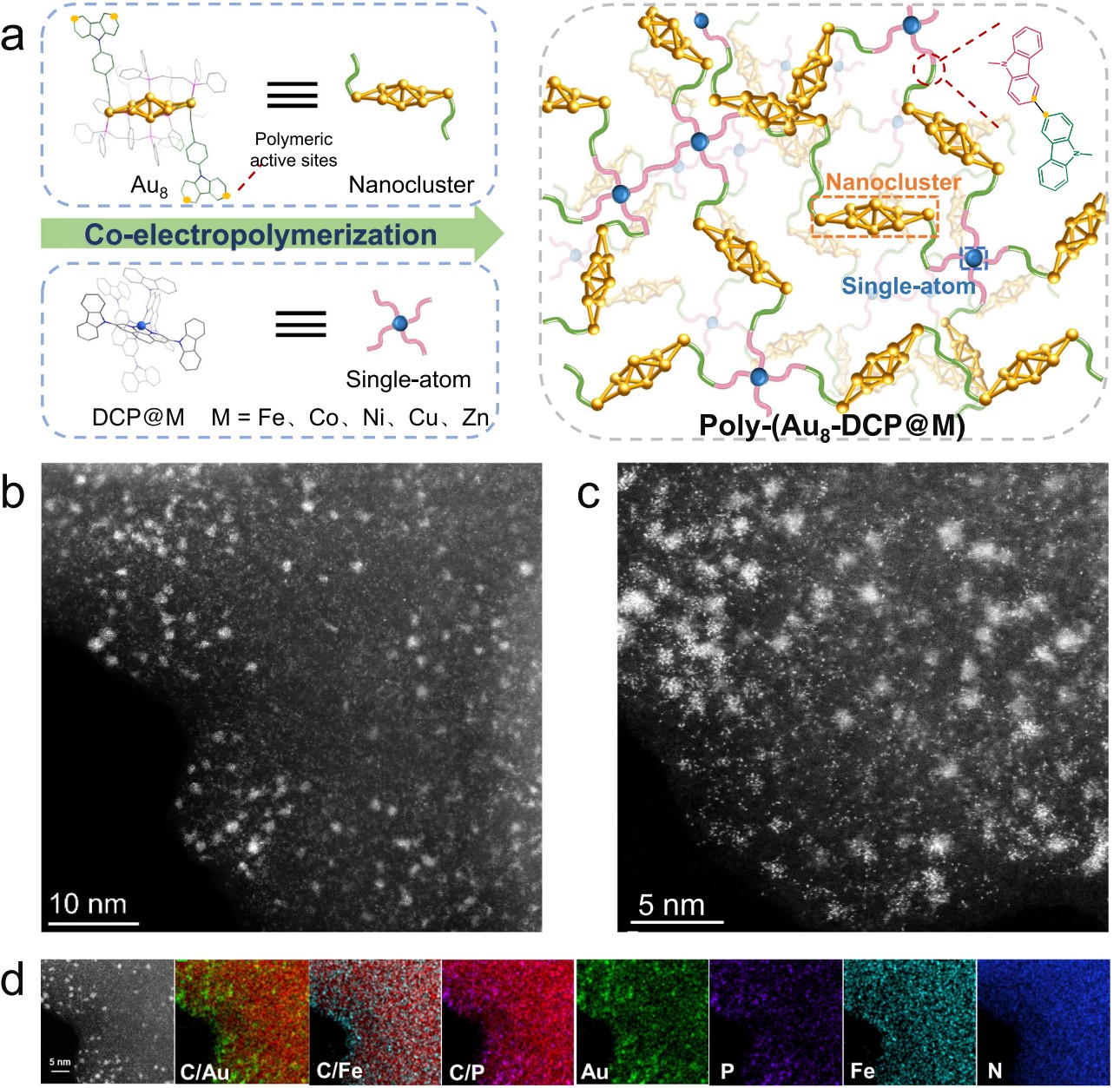

**Fig. 1 | Synthesis procedure and morphology characterizations of the catalysts. a** Schematic illustration of Poly-(Au$_8$-DCP@M) catalysts fabricated by co-eletropolymerization strategy. **b**, **c** Atomic-resolution HAADF-STEM image of Poly-(Au$_8$-DCP@Fe) revealing the atomically dispersed Au$_8$ NCs and Fe SA. **d** Elemental mapping of overlapped images and Au (green), P (purple), Fe (cyan), and N (blue).

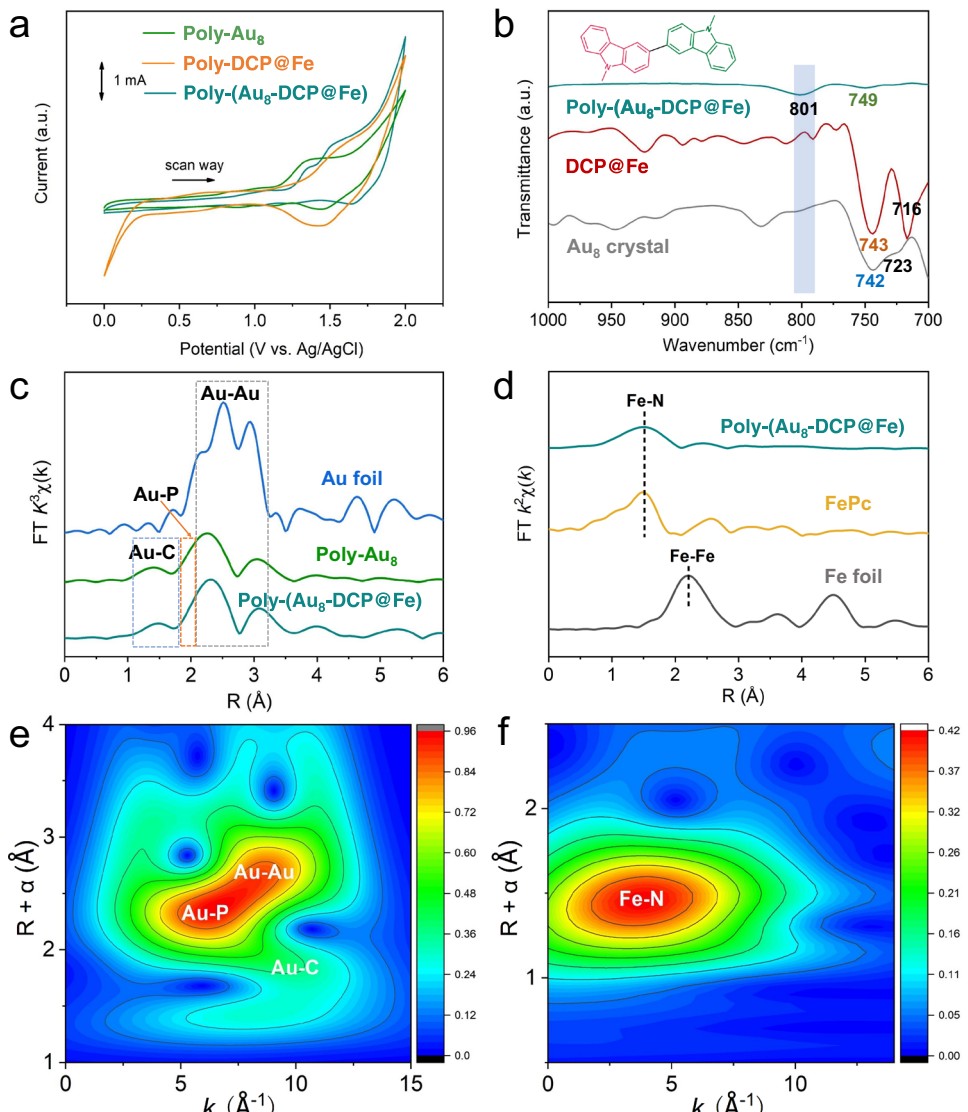

**Fig. 2 | Structural analysis of Poly-(Au₈-DCP@Fe). a** CV profiles of co-eletropolymerization recorded for first scan cycles at 100 mV s⁻¹ (the electro-polymerization potential range was set between 0 V and 2.0 V. Carbon paper, Ag/Ag⁺ and platinum foil were used as the working electrode, reference electrode and counter electrode, respectively). **b** FT-IR spectra of the Poly-(Au₈-DCP@Fe), DCP@Fe, and Au₈ crystal, as well as their peak assignments. **c** Au L₃-edge EXAFS of Au foil, Poly-Au₈, and Poly-(Au₈-DCP@Fe). **d** Fe K-edge EXAFS of Poly-(Au₈-DCP@Fe), FePc, and Fe foil. Wavelet transform of (**e**) Au L₃-edge and (**f**) Fe K-edge EXAFS for Poly-(Au₈-DCP@Fe).

## Results

### Synthesis and characterization of nanocluster-single atom catalysts

The preparation of Poly-(Au₈-DCP@M) is schematically depicted in Fig. 1a. Specifically, alkynyl carbazole ligand (HCzPA) modified Au₈ nanoclusters were successfully synthesized via solvent evaporation method. Supplementary Fig. 3 and Supplementary Tables 1–3 shows its identical atomic structure solved by single-crystal X-ray diffraction (SCXRD). The Au₈ core contained a di-edge-bridged bitetrahedral unit. Each of the two terminal gold atoms accommodates one phosphine and one alkynyl group. The highly electro-active group "carbazole" in the CzPAs endow Au₈ clusters excellent polymerization capability. The bulk phase purity of this crystal was verified by powder X-ray diffraction (PXRD) (Supplementary Fig. 4). Another carbazole-substituted phenanthroline monomer chelated with different single site metals (DCP@M, M = Fe, Co, Ni, Cu, Zn) was successively electrochemically polymerized with Au₈ crystal via multi-cycled cyclic voltammetry(CV) methods in the potential range of 0 to 2 V (vs. Ag/Ag⁺) to obtain Poly-(Au₈-DCP@M). For comparison, a series of control samples, including

co-polymerization of Au₈ and DCP (Poly-(Au₈-DCP)), as well as self-polymerization of monomers Au₈ (Poly-Au₈) and DCP@M (Poly-DCP@M) were prepared via the same procedure (Supplementary Fig. 5). Taking Poly-(Au₈-DCP@Fe) as a representative sample, we conducted a series of characterizations. In the first cycle of the positive CV scan, Poly-(Au₈-DCP@Fe) displayed two obvious oxidation peaks at 1.36 V and 1.50 V, which are attributed to the oxidation of carbazole groups in both the Au₈ crystal and DCP@Fe. The reduction peak of Poly-(Au₈-DCP@Fe) appeared at 1.68 V in the negative CV scan cycle is completely different from that of Au₈ (1.44 V) and DCP@Fe (1.42 V), suggesting that the new dimeric carbazole cations formed by carbazole cations coupling in Au₈ and DCP@Fe were reduced (Fig. 2a). In the FT-IR spectra, the generation of new peaks at 801 cm⁻¹ was attributed to the vibrational bands of C−H bonds of the tri-substituted carbazole ring as well as the disappearance of the di-substituted carbazole ring at 716 and 723 cm⁻¹ (Fig. 2b), thus indicating the formation of the bis-carbazole network[40–42].

High-resolution transmission electron microscopy (TEM) image confirmed that the Au₈ NCs were uniformly dispersed in the Poly-(Au₈-

DCP@Fe) (Supplementary Fig. 6). The co-existence of metal NCs and single metal sites on Poly-($Au_8$-DCP@Fe) was confirmed by atomic-resolution high-angle annular dark-field scanning TEM (HAADF-STEM). Figure 1b, c clearly showed the highly dispersed dots with heavier image contrast, thus showing the presence of ultrafine $Au_8$ clusters. The size (~1.0 nm) is close to the X-ray crystallographically measured value of an $Au_8$ nanocluster (0.98 nm). This indicates that this strategy efficiently prevents cluster agglomeration. Closer observation further shows that numerous single metal sites were homogeneously presented on the hybrid material. Energy dispersive X-ray spectroscopy (EDS) mapping analysis of STEM images confirmed that the Fe single sites were homogeneously distributed over Poly-($Au_8$-DCP@Fe), while the specific elements (Au and P) of $Au_8$ were uniformly dispersed on Poly-($Au_8$-DCP@Fe) in the form of clusters (Fig. 1d). As a control sample, TEM analysis and HAADF-STEM imaging of Poly-DCP@Fe confirmed the absence of metal nanoparticles, showcasing the uniform dispersion of atomically dispersed Fe species (Supplementary Figs. 7, 8). Quantitative analysis of Fe and Au contents in Poly-($Au_8$-DCP@Fe) and other control samples ($Au_8$ crystal, Poly-$Au_8$, Poly-($Au_8$-DCP), DCP@Fe and Poly-DCP@Fe) were determined via inductively coupled plasma optical emission spectrometer (ICP-OES) (Supplementary Table 4). The result revealed Au and Fe contents of 5.99 wt% and 4.00 wt%, respectively, in Poly-($Au_8$-DCP@Fe).

Furthermore, the local coordination environment and chemical states of Fe and Au species in Poly-($Au_8$-DCP@Fe) were investigated via X-ray absorption spectroscopy. The Au $L_3$-edge X-ray absorption near-edge structure (XANES) results showed that the white lines of Poly-$Au_8$, Poly-($Au_8$-DCP), and Poly-($Au_8$-DCP@Fe) are very close to each other, indicating that the average oxidation state of the cluster is close to a metallic state. The near-edge of the Au $L_3$-edge spectra for Poly-($Au_8$-DCP@Fe) displayed a discernible shift towards the high-energy region compared to that of Poly-($Au_8$-DCP), indicating an elevated oxidation state of Au atoms in Poly-($Au_8$-DCP@Fe) following the introduction of Fe single atoms (Supplementary Fig. 9). This observation aligns with the findings obtained from X-ray photoelectron spectroscopy (XPS) analysis. Fourier transformed (FT) extended X-ray adsorption fine structure (EXAFS) spectra showed a peak at around 1.5 Å in the R space for Poly-$Au_8$ and Poly-($Au_8$-DCP@Fe), which was ascribed to the Au−C bond. The broad peaks appearing in the range of 1.8–3.2 Å were attributed to Au−P and Au−Au bonds (Fig. 2c and Supplementary Table 5). Wavelet transform (WT) EXAFS plots in K space further confirm the presence of Au−Au, Au−P, and Au−C bond in the Poly-($Au_8$-DCP@Fe) and Poly-$Au_8$ (Fig. 2e and Supplementary Fig 10). These results collaborated that the pristine coordination bond of $Au_8$ NCs after polymerization is well-preserved. The Fe $K$-edge XANES spectra showed that the absorption edges of Poly-($Au_8$-DCP@Fe) is close to $Fe_3O_4$, indicating that the valence states of Fe in the Poly-($Au_8$-DCP@Fe) was between +2 and +3 (Supplementary Fig. 11). The EXAFS spectrum of Fe $K$-edge for Poly-($Au_8$-DCP@Fe) in R space displays only a major peak identified at 1.50 Å. The position of this peak is analogous to that of iron phthalocyanine (FePc) references, signifying the existence of Fe-N coordination. The absence of Fe-Fe bonding in Poly-($Au_8$-DCP@Fe) verified the atomic distribution of the Fe species (Fig. 2d). WT EXAFS plots also demonstrate the existence of Fe SAs configurations in catalysts (Fig. 2f and Supplementary Fig. 12). These results demonstrated the successful fabrication of $Au_8$ NC and Fe SA hybrid material.

To explore the interaction between $Au_8$ clusters and Fe single sites, the electronic states of Au species in Poly-($Au_8$-DCP@Fe) were studied by XPS. Compared to Poly-($Au_8$-DCP), the Au $4f_{7/2}$ of Poly-($Au_8$-DCP@Fe) shift to higher binding energy, indicating an increasing $Au^{\delta+}$ species after the introduction of Fe sites. The high oxidation state of Au species could promote the stabilization of reaction intermediates *COOH and *CO, which could make the transformation from $CO_2$ to the *COOH intermediate more favorable in $CO_2$RR[43]. Similarly, the Au $4f_{7/2}$

binding energy of Poly-$Au_8$ was 85.48 eV, which was about 0.48 eV higher than the binding energy of $Au_8$ crystal (Fig. 3a). For the Fe charge state, the negative shift of the binding energies of Fe $2p$ in Poly-($Au_8$-DCP@Fe) than that of Poly-DCP@Fe indicated an decreasing valance state of Fe SA (Supplementary Fig. 13). The change of the electronic state of Poly-($Au_8$-DCP@Fe), Poly-($Au_8$-DCP) and Poly-(DCP@Fe) was further probed by diffuse reflectance infrared Fourier transform spectroscopy under a CO atmosphere (CO-DRIFTS). The adsorbed CO peak position is considered as a cue for determining the electron cloud density of accessible Au site[44–47]. As shown in Fig. 3b, compared with Poly-($Au_8$-DCP), a red-shift of CO molecules adsorbed onto the $Au_8$ NCs is observed for Poly-($Au_8$-DCP@Fe), suggesting a decrease in electron density and increasing valance state of $Au_8$ NCs. In contrast, blue-shift occurred for Fe sites in Poly-($Au_8$-DCP@Fe) compared to the peak of Poly-DCP@Fe, suggesting an increase in the electron density of Fe SA. The peak at around 2150–2250 $cm^{-1}$ was assigned to gaseous CO[45,47,48].

To further reveal the electron redistribution behavior, charge density difference and Bader charge analyses were performed. The relaxed atomic structural models of $Au_8$ crystal, Poly-$Au_8$, Poly-($Au_8$-DCP), and Poly-($Au_8$-DCP@Fe) were constructed based on the structural analysis. Once the Fe SA site was introduced into the hybrid system, the electron around the $Au_8$ cluster active site transfers to the peripheral ligands, thus making the $Au_8$ cluster center in Poly-($Au_8$-DCP@Fe) more positive and strengthening the adsorption of the intermediate in $CO_2$RR (Fig. 3c). In addition, the Bader charge for $Au_8$ cluster of Poly-($Au_8$-DCP) and Poly-($Au_8$-DCP@Fe) was calculated to be 0.126 and 0.101, respectively. Compared to Poly-($Au_8$-DCP), the electron cloud density of $Au_8$ in Poly-($Au_8$-DCP@Fe) towards the electron-deficient state, which is consistent with the XPS results. The partial density of states (PDOS) diagrams in Fig. 3d illustrate that the d-band center of Poly-($Au_8$-DCP@Fe) (−3.39 eV) was much closer to the Fermi level than that of $Au_8$ crystal (−4.19 eV), Poly-$Au_8$ (−4.44 eV) and Poly-($Au_8$-DCP) (−3.85 eV), thus indicating that the introduction of SAs can result in more stable adsorption with the intermediate. This optimizes the catalytic performance of $Au_8$ NCs. The above analysis indicated that the functionalizing SAs induces the electron redistribution around the $Au_8$ NCs, which modulates the electronic structure of $Au_8$ NCs and therefore greatly influences their inherent catalytic performance.

## Electrocatalytic activity study of Poly-($Au_8$-DCP@Fe): electrocatalytic $CO_2$ reduction

Encouraged by the regulated electronic state of $Au_8$ cluster, the electrochemical $CO_2$RR catalytic performances of Poly-($Au_8$-DCP@M) were next examined in 0.5 M $KHCO_3$ using a three-electrode H-type cell. The gaseous and liquid products were determined by gas chromatography (GC) and $^1$H nuclear magnetic resonance (NMR) spectroscopy, respectively. No liquid products were found, and the gas products were CO and $H_2$ for all samples (Supplementary Fig. 14). Figure 4a and Supplementary Fig. 15 show that the maximum CO Faradaic efficiency (FE) of Poly-($Au_8$-DCP@M) was 90.89% (Fe, −0.57 V), 87.23% (Co, −0.67 V), 74.14% (Ni, −0.57 V), 50.01% (Cu, −0.47 V) and 9.25% (Zn, −0.57 V), respectively. Of these, Poly-($Au_8$-DCP@Fe) exhibited highest $CO_2$RR performance. Although Co functionalized catalyst Poly-($Au_8$-DCP@Co) exhibited high comparable CO FE, more negative reduction potentials implied more energy required for the catalytic reactions to proceed. Then, a series of control experiments were conducted to understand the origination of high activity of Poly-($Au_8$-DCP@Fe). Linear scanning voltammetry (LSV) curves showed a greater current density and a more positive onset potential of Poly-($Au_8$-DCP@Fe) compared to the other four control catalysts: $Au_8$ crystal, Poly-($Au_8$-DCP), Poly-$Au_8$, and Poly-DCP@Fe (Fig. 4b and Supplementary Fig. 16). Moreover, at −0.57 V, Poly-($Au_8$-DCP@Fe) exhibited a higher CO FE (90.89%) than $Au_8$ crystal (0.60%), Poly-

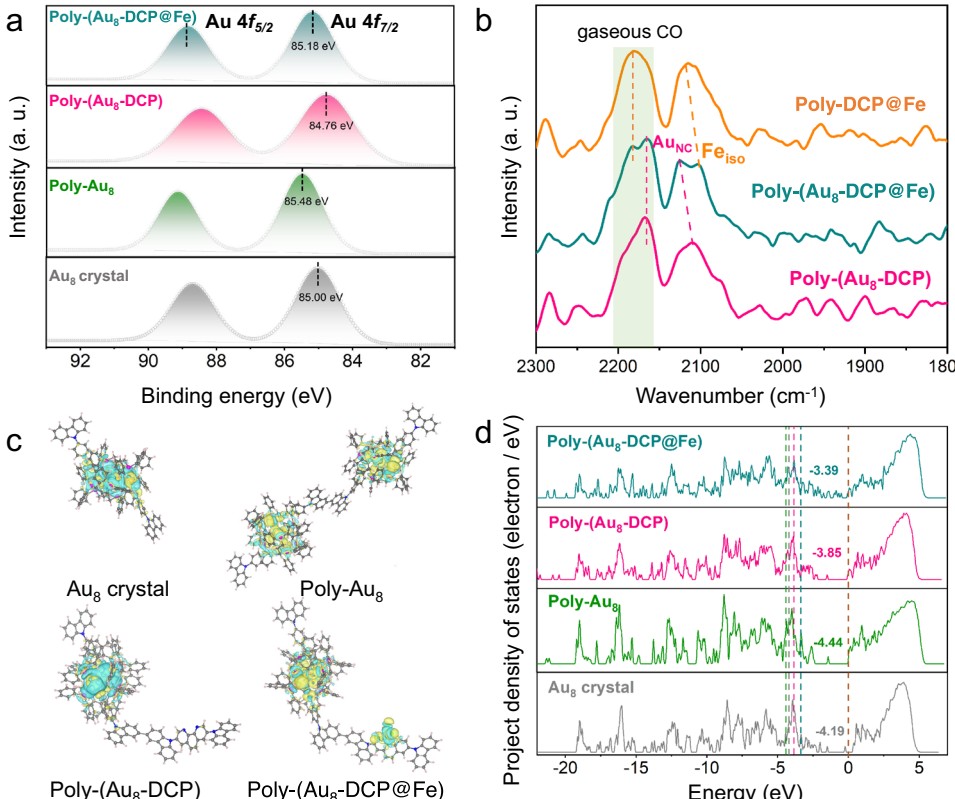

**Fig. 3 | Study on electronic structure of Poly-(Au₈-DCP@Fe). a** XPS spectra of Au $4f$ for Poly-(Au₈-DCP@Fe), Poly-(Au₈-DCP), Poly-Au₈, and Au₈ crystal. **b** Spectra of CO-DRIFTS on Poly-(Au₈-DCP@Fe), Poly-(Au₈-DCP), and Poly-DCP@Fe at 298 K. **c** Differential charge density distributions of Au₈ crystal, Poly-Au₈, Poly-(Au₈-DCP), and Poly-(Au₈-DCP@Fe). The yellow and blue regions correspond to the charge accumulation and depletion, respectively. **d** The partial density of states (PDOS) for Au₈ crystal, Poly-Au₈, Poly-(Au₈-DCP), and Poly-(Au₈-DCP@Fe).

Au₈ (12.56%), Poly-(Au₈-DCP) (69.06%) and Poly-DCP@Fe (1.16%) (Fig. 4c and Supplementary Figs. 17–21). The remarkable enhancement in CO FE of Poly-Au₈ beyond −0.57 V is attributed to a sharp increase in current density[49,50] (Supplementary Fig. 22). *Quasi-in* situ XPS analysis further indicated that the enhanced performance post −0.57 V is influenced by the partial exposure of catalytic sites of Poly-Au₈ (Supplementary Fig. 23). Additionally, the catalytic performance evaluation of DCP@Fe, as shown in Supplementary Fig. 24, demonstrates the production of H₂ as the primary gas in a broad potential range from −0.47 to −0.87 V, with H₂ FE near 100%, mirroring the behavior of Poly-DCP@Fe. These findings collectively indicated that Fe SAs in these catalytic systems are inactive in the electrocatalytic CO₂ to CO reaction. The higher activity of Poly-(Au₈-DCP@Fe) than that of Poly-(Au₈-DCP) between −0.47 to −0.87 V indicated that the introduction of single Fe sites promotes the CO₂RR. Interestingly, Poly-(Au₈-DCP) showed significantly increased catalytic efficiency than isolated Au₈ cluster and Poly-Au₈. This may be due to the introduction of the DCP component to promote electron and mass transfer of Poly-(Au₈-DCP). Moreover, the poorer catalytic performance of Poly-DCP@Fe than of Poly-Au₈ suggests that Au₈ NC rather than Fe SA is likely the main active site. To substantiate this inference, 4,4′-Di(9H-carbazol-9-yl)-1,1′-biphenyl (BCP), featuring two carbazole groups akin to Au₈ clusters, was employed as a comonomer for dispersing Fe SAs by co-eletropolymerization with DCP@Fe (Supplementary Figs. 25 and 26). The obtained Poly-(DCP@Fe-BCP) predominantly produces H₂ as the main gas across a broad potential range from −0.47 to −0.87 V, with H₂ FE near 100% (Supplementary Fig. 25b). On the contrary, Poly-(Au₈-BCP), prepared by co-electropolymerization of Au₈ clusters with BCP, exhibited significantly higher CO₂RR activity (CO FE$_{maximum}$ = 78.32% at −0.57 V) (Supplementary Fig. 25c and

Supplementary Fig. 27). Taken together, these results underscore that sufficiently dispersed Au₈ cluster, rather than Fe SA, are more likely to serve as catalytic active sites, driving the catalyst's activity for CO₂-to-CO conversion.

To estimate the electrochemical active surface areas (ECSAs) of these samples and further discuss the potential influencing factors, we calculated the electrochemical double-layer capacitance ($C_{dl}$). The $C_{dl}$ values of these samples decreased in the following order: Poly-(Au₈-DCP@Fe) (0.275 mF cm⁻²) > Poly-(Au₈-DCP) (0.271 mF cm⁻²) > Poly-Au₈ (0.197 mF cm⁻²) > Au₈ crystal (0.01 mF cm⁻²). After polymerization, the larger ECSA of Poly-(Au₈-DCP@Fe), Poly-(Au₈-DCP), and Poly-Au₈ than that of pristine Au₈ cluster implied more accessible active sites for CO₂RR. These differences originated from the specific polymerized network building the high electron transfer pathway to the catalytic sites. Since the ECSA of Poly-(Au₈-DCP@Fe) is similar to that of Poly-(Au₈-DCP), but its current density is much larger, thus demonstrating the much better intrinsic high activity of Poly-(Au₈-DCP@Fe) (Fig. 4d and Supplementary Fig. 28). Besides, the Tafel slopes were calculated from a characteristic curve of the overpotential versus a logarithm of the steady partial current density of CO. The Tafel slope value of Poly-(Au₈-DCP@Fe) was 218 mV decade⁻¹, which is successively lower than that of Poly-(Au₈-DCP) (237 mV decade⁻¹), Poly-Au₈ (240 mV decade⁻¹) and Au₈ crystal (381 mV decade⁻¹), thus illustrating its faster CO₂RR kinetics (Supplementary Fig. 29). In short, the enhanced intrinsic catalytic activity and kinetics of Poly-(Au₈-DCP@Fe) fully demonstrate the activation effect of the decorating isolated Fe atoms on the Au₈ cluster for enhanced CO₂RR activity. The long-term CO₂RR catalytic stability of Poly-(Au₈-DCP@Fe) was studied at a fixed potential of −0.57 V. The corresponding CO FE can be retained at values above 80% over the entire experiment, and the current density

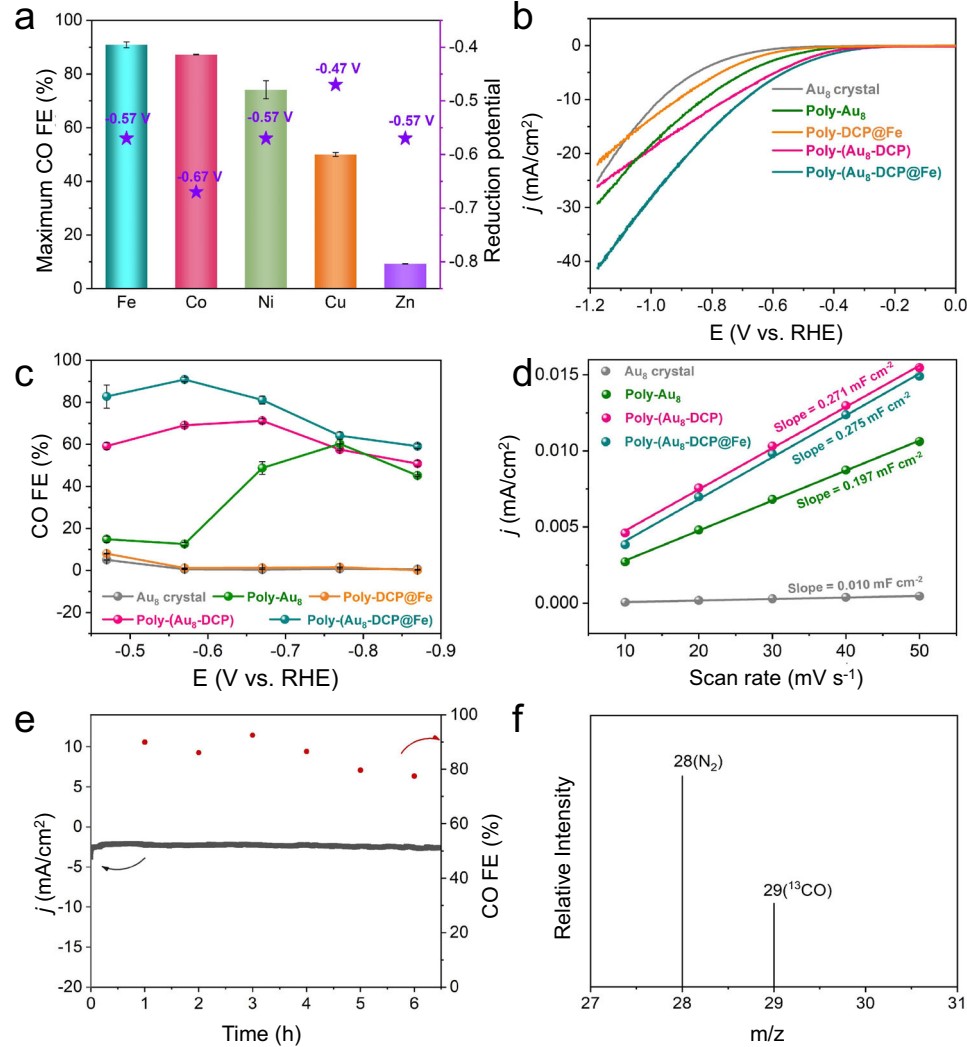

**Fig. 4 | Electrocatalytic $CO_2$ reduction performance. a** The maximum CO FE of Poly-(Au$_8$-DCP@M). **b** The LSV curves of Au$_8$ crystal, Poly-(Au$_8$-DCP@Fe), Poly-(Au$_8$-DCP), Poly-Au$_8$, and Poly-DCP@Fe measured in $CO_2$-saturated 0.5 M KHCO$_3$, (**c**) CO FE of Poly-(Au$_8$-DCP@Fe), Poly-(Au$_8$-DCP), Poly-Au$_8$, Au$_8$ crystal, and Poly-DCP@Fe. **d** Plots of the current density versus the scan rate for Poly-(Au$_8$-DCP@Fe), Poly-(Au$_8$-DCP), Poly-Au$_8$ and Au$_8$ crystal, the linear fit determined the specific capacitance. **e** Chronoamperometry of the Poly-(Au$_8$-DCP@Fe) in $CO_2$ saturated 0.5 M KHCO$_3$ at −0.57 V. **f** GC-MS of $^{13}CO$ generation using Poly-(Au$_8$-DCP@Fe) as electrocatalyst in $CO_2$RR reaction under $^{13}CO_2$ atmosphere.

also remained stable in the later stage after 6 h of testing (Fig. 4e). The marginal degradation in CO FE observed in Poly-(Au$_8$-DCP@Fe) can be primarily ascribed to the formation of a limited number of larger-sized Au$_8$ nanoclusters after the long-term reaction (Supplementary Fig. 30). The structure of post-test Poly-(Au$_8$-DCP@Fe) was confirmed by HAADF-STEM and XPS (Supplementary Figs. 31 and 32). The well-maintained morphology and electronic states underscored its high stability. To identify the carbon source of the reduction products, $^{13}CO_2$ isotopic experiment was carried out. The peak at m/z = 29 assigned to $^{13}CO$ indicates that the reduction product was indeed from the reactant $CO_2$ (Fig. 4f). In comparison with recently reported Au NCs-based catalysts, such as Au$_{22}$H$_3$ (CO FE$_{maximum}$ = 92.7% at −0.60 V)[51] and Au$_{24}$H$_3$ (CO FE$_{maximum}$ > 90% at 10 mA/cm$^{-2}$)[52], Poly-(Au$_8$-DCP@Fe) (CO FE$_{maximum}$ = 90.89% at −0.57 V) demonstrated comparable $CO_2$RR performance in terms of CO FE and the applied potential. A comprehensive comparison, presented in Supplementary Fig. 33 and Supplementary Table 6, showcased the $CO_2$RR performance of Poly-(Au$_8$-DCP@Fe) against reported Au NCs and bimetallic Au NCs-based catalysts, underscoring the exceptional activity of Poly-(Au$_8$-DCP@Fe).

## In situ characterization and theoretical analysis of intermediates for the overall process of $CO_2$RR

To probe possible intermediates and reaction pathways during $CO_2$RR, in situ attenuated total reflection Fourier transform infrared (ATR-FTIR) spectroscopy of Poly-Au$_8$, Poly-(Au$_8$-DCP) and Poly-(Au$_8$-DCP@Fe) at different application potentials were recorded, respectively. As shown in Fig. 5a−c, all samples exhibited the characteristic peaks of *COOH signal located at 1400 cm$^{-1}$. Notably, the intensity of *COOH peaks of Poly-(Au$_8$-DCP) and Poly-(Au$_8$-DCP@Fe) was obviously stronger than that of Poly-Au$_8$, thus suggesting the facilitated *COOH formation on these two materials. The peaks ranging from 1800 to 1900 cm$^{-1}$ correspond to the bridge-adsorbed *CO (*CO$_B$) on Au surfaces. Obviously, Poly-(Au$_8$-DCP@Fe) displayed a stronger *CO$_B$ band intensity than Poly-(Au$_8$-DCP) and Poly-Au$_8$. The red shift of such peaks for Poly-(Au$_8$-DCP@Fe) at more negative potentials is caused by the Stark tuning effect[53,54]. In addition, Poly-(Au$_8$-DCP@Fe) showed an additional absorption peak at 2000-2100 cm$^{-1}$ owing to the linear-adsorbed *CO (*CO$_L$) on Fe SA. These results confirmed that Poly-(Au$_8$-DCP@Fe) is more efficient in converting $CO_2$ into CO than Poly-(Au$_8$-DCP) and Poly-Au$_8$.

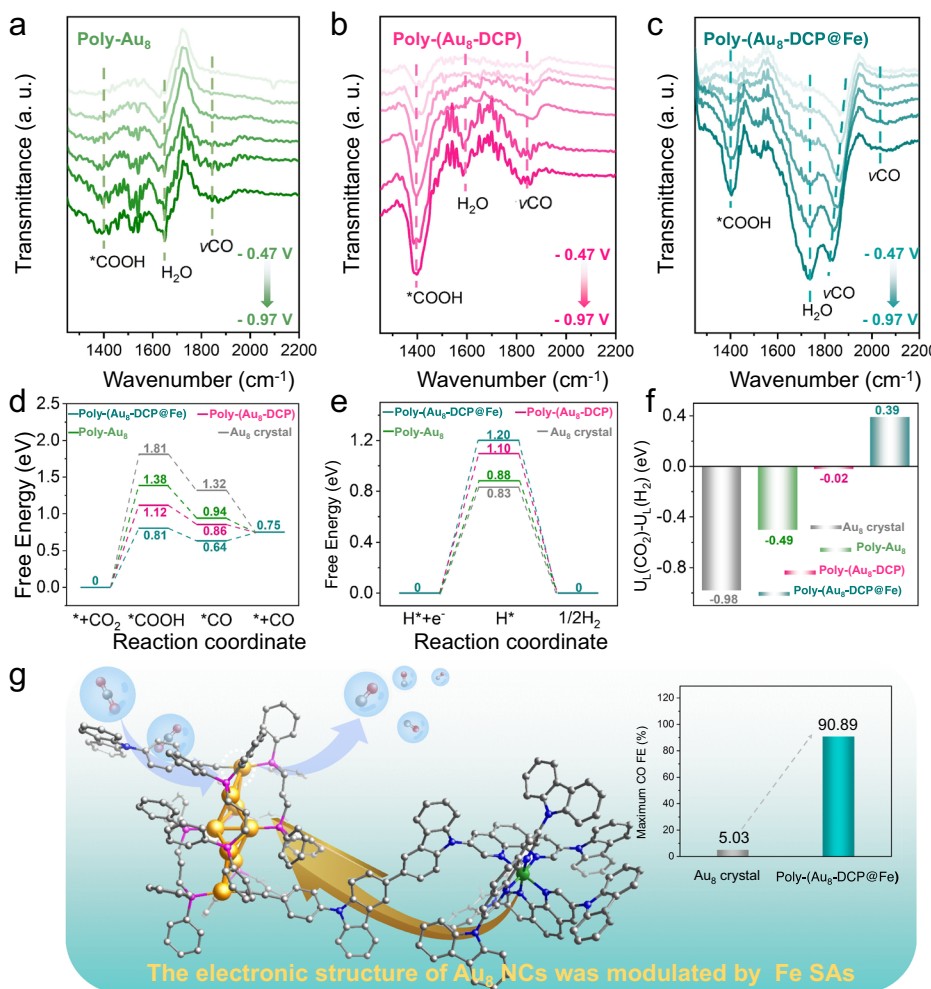

**Fig. 5 | In situ ATR-FTIR and theoretical studies.** In situ ATR-FTIR spectra of (**a**) Poly-Au$_8$, (**b**) Poly-(Au$_8$-DCP) and (**c**) Poly-(Au$_8$-DCP@Fe) in CO$_2$ saturated 0.5 M KHCO$_3$ between −0.47 V and −0.97 V. The free energy diagram of the (**d**) CO$_2$RR and (**e**) HER over Au$_8$ crystal, Poly-Au$_8$, Poly-(Au$_8$-DCP), and Poly-(Au$_8$-DCP@Fe). **f** The values of U$_L$(CO$_2$)-U$_L$(H$_2$) for Au$_8$ crystal, Poly-Au$_8$, Poly-(Au$_8$-DCP), and Poly-(Au$_8$-DCP@Fe). **g** Conceptual illustration of the electronic structure of Au$_8$ cluster modulated by SA in Poly-(Au$_8$-DCP@Fe).

Density functional theory (DFT) calculations were carried out to further shed light on the critical role of Fe SAs in boosting the CO$_2$RR activity of Au$_8$ clusters. Figure 5d, e depicts the reaction-free energy pathway for the electrochemical CO$_2$RR and competitive side hydrogen evolution reaction (HER) over four materials. The Au$_8$, Poly-Au$_8$, Poly-(Au$_8$-DCP), and Poly-(Au$_8$-DCP@Fe) displayed the required free energy changes (ΔG) of 1.81, 1.38, 1.12, and 0.81 eV, respectively, for the rate-limiting step of reduction of CO$_2$ to *COOH. These calculations suggest that Poly-(Au$_8$-DCP@Fe) had a more optimal ΔG, thus leading to a lower barrier forming *COOH, from the thermodynamic point. Moreover, the energy barriers of HER are significantly increasing for Poly-(Au$_8$-DCP@Fe), which inhibits the production of H$_2$ (Fig. 5e). In addition, the higher energy barrier for electrochemical CO$_2$RR and the lower energy barrier for HER exhibited by Fe SA sites than Au$_8$ NCs sites in Poly-(Au$_8$-DCP@Fe), indicate that the Au$_8$ NC sites in Poly-(Au$_8$-DCP@Fe) were more responsible for the high catalytic efficiency of CO$_2$RR (Supplementary Fig. 34). We also calculated the free energy of CO$_2$RR and HER for catalysts chelating different single atoms (M = Co, Ni, Cu, Zn), and all catalysts showed a consistent trend, mirroring that observed in Poly-(Au$_8$-DCP@Fe) (Supplementary Fig. 35). The corresponding reaction pathway and the free energy diagram of each intermediate over the Au$_8$ NC sites are shown in Supplementary Fig. 36. Synthetically, the difference between thermodynamic limiting potentials for the CO$_2$RR and HER (U$_L$(CO$_2$)−U$_L$(H$_2$)) reflects the

selectivity for CO$_2$-to-CO. It is shown that Poly-(Au$_8$-DCP@Fe) present more positive value (0.39 eV) among all catalysts, explaining its highest CO$_2$RR selectivity (Fig. 5f). These are identical to the experimental results. Therefore, the experimental and theoretical evidences indicate regulate the electronic structure of NCs by introducing extra SAs is a feasible method to improve the inherent catalytic performance of NCs (Fig. 5g). Electrostatic potential (ESP) analysis of Poly-(Au$_8$-DCP@Fe) revealed negative ESP regions around the organic moieties of surface ligands, fostering robust interactions with CO$_2$ molecules and elevating CO$_2$ concentration near active sites[55,56]. Furthermore, DFT calculations showed that *COOH intermediate adsorption site on Poly-(Au$_8$-DCP@Fe) to be No. 9 Au atom, suggesting that the catalytic site for CO$_2$ reduction predominantly resides at this specific location (Supplementary Fig. 37).

## Discussion
In summary, we have successfully constructed a highly homogeneous NC-SA hybrid catalyst Poly-(Au$_8$-DCP@Fe) by facile co-eletropolymerization strategy. The obtained Poly-(Au$_8$-DCP@Fe) featured a considerably enhanced catalytic performance toward CO$_2$RR— a 90.89% CO FE at −0.57 V, than that of SA or metal NC components. The detailed analysis revealed that the high catalytic activity of Poly-(Au$_8$-DCP@Fe) derive from the effective regulation of the electronic structure of the Au$_8$ NCs by the Fe SAs, in which the Fe SA showed a

positive modulation on the charge density and projected density of states of the metal cluster sites. This work creatively presented a new strategy to regulate the inherent catalytic activity of metal NCs via the introduction of SAs.

## Methods

### Synthesis of ligand 9-(4-ethynylphenyl)carbazole (HCzPA)

The ligand 9-(4-ethynylphenyl)carbazole (HCzPA) was synthesized according to the previous report[57]. $^1$H NMR of HCzPA. (600 MHz, CDCl$_3$): δ 8.16–8.12 (m, 2H), 7.75–7.71 (m, 2H), 7.58–7.51 (m, 2H), 7.46–7.38 (m, 4H), 7.34–7.26 (m, 2H), 3.18 (s, 1H).

### Synthesis of Au$_8$ crystal

The taget [Au$_8$(dppp)$_4$(CzPA)$_2$]$^{2+}$ clsuters were synthesized using [Au$_8$(dppp)$_4$](NO$_3$)$_2$[58] as precursor. A methanolic solution (50 mL) of [Au$_8$(dppp)$_4$](NO$_3$)$_2$ (30.0 mg, 9 μmol) was added to HCzPA (5.0 mg, 18 μmol) and sodium methoxide (145 mg, 2.7 mmol), and the mixture was stirred at room temperature for 10 h. The obtained mixture was treated with water and then extracted with dichloromethane (20 mL × 3). The combined organic phase was dried over anhydrous Na$_2$SO$_4$, filtered and evaporated to dryness to give a pinkish solid, which was further purified by vapor diffusion of ether into a cluster solution in dichloromethane to give Au$_8$ crystal as red crystals.

### Synthesis of 3,8-di(9H-carbazol-9-yl)-1,10-phenanthroline (DCP)

3,8-Di(9H-carbazol-9-yl)-1,10-phenanthroline was prepared according to literature procedures[59]. $^1$H NMR of DCP. (600 MHz, DMSO): δ 9.45 (s, 2H,), 8.98 (s, 2H), 8.35 (d, 4H, J = 8.0 Hz), 8.28 (s, 2H), 7.61 (d, 4H, J = 8.0 Hz), 7.53 (t, 4H, J = 8.0 Hz), 7.39 (t, 4H, J = 8.0 Hz).

### Synthesis of DCP@M (M═Fe, Co, Ni, Cu, Zn)

DCP (30 mg, 0.06 mmol) was refluxed with FeCl$_3$ (9.54 mg, 0.06 mmol) in *N,N*-dimethylformamide under argon overnight. After cooling to room temperature, the mixture was evaporated to dryness and washed with water to give a red solid (DCP@Fe).

The DCP@Co, DCP@Ni, DCP@Cu and DCP@Zn were similarly prepared in methanol following the procedure for DCP@Fe by using Co(NO$_3$)$_2$·6H$_2$O, NiCl$_2$·6H$_2$O, Cu(NO$_3$)$_2$·3H$_2$O and ZnCl$_2$, respectively.

### Preparation of Poly-(Au$_8$-DCP@M) (M═Fe, Co, Ni, Cu, Zn)

The preparation of Poly-(Au$_8$-DCP@Fe) was given as a typical example for the preparation of Poly-(Au$_8$-DCP@M). The co-electropolymerization of Au$_8$ crystal and DCP@Fe was performed using a standard three-electrode system attached to an CH660E B14145 Electrochemical Workstation. Platinum foil (Pt) was used as the counter electrode, carbon paper/indium-doped tin oxide (ITO) glass as the working electrode, and Ag/AgCl electrode as the reference electrode. Co-eletropolymerization of Au$_8$ crystal and DCP@Fe was carried out in CH$_3$CN/CH$_2$Cl$_2$ (2/1 *v/v*) by using 0.1 M LiClO$_4$ as electrolyte via multi-cycled cyclic voltammetry (CV) methods in the potential range of 0 to 2 V, and the scan rate was 100 mV s$^{-1}$. After the electropolymerization process (20 cycles), the resulting Poly-(Au$_8$-DCP@Fe) was washed with CH$_2$Cl$_2$ to remove unreacted precursors.

### Preparation of Poly-(Au$_8$-DCP), Poly-Au$_8$, and Poly-DCP@Fe

Poly-(Au$_8$-DCP) was similarly prepared following the procedure for Poly-(Au$_8$-DCP@M) by using Au$_8$ crystal and DCP. Poly-Au$_8$ and Poly-DCP@Fe were fabricated by self-polymerization of Au$_8$ crystal and DCP@Fe, which similarly to the procedure for Poly-(Au$_8$-DCP@M).

### Evaluation of electrochemical CO$_2$ reduction (CO$_2$RR) performance

The electrochemical CO$_2$RR was carried out using a CH660E B14145 electrochemical workstation in an H-type electrolytic cell separated by a proton membrane. The Poly-(Au$_8$-DCP@M)/Poly-(Au$_8$-DCP)/Poly-

Au$_8$/Poly-DCP@Fe/Au$_8$ crystal was used as the working electrode, which had a controlled surface area of 1 cm$^2$. The Ag/AgCl (saturated with KCl) and platinum foil (Pt) were used as the reference electrode and counter electrode, respectively. 0.5 M KHCO$_3$ solution (pH = 7.3) was evenly distributed to the cathode and anode compartments. CO$_2$ gas was delivered at an average rate of 20 mL min$^{-1}$ (at room temperature and ambient pressure), and the separated gas products were analyzed by Agilent GC7820 Gas Chromatograph. The liquid products were analyzed afterwards by quantitative NMR (Bruker AVIII HD 600) All the potentials in this study were converted to values with respect to a reversible hydrogen electrode (RHE) by E (vs. RHE) = E (vs. Ag/AgCl) + 0.0591 × pH +0.197.

The Poly-(Au$_8$-DCP@M), Poly-(Au$_8$-DCP), Poly-Au$_8$, and Poly-DCP@Fe for CO$_2$RR were deposited on carbon paper as work electrode. As a control, catalyst inks were prepared by dissolving 1 mg Au$_8$ crystal into 0.5 mL solution including 0.46 mL dichloromethane and 40 μL Nafion solution, dripped on the carbon paper (1 × 1 cm$^{-2}$) and then dried at room temperature.

The faradaic efficiency (FE) for CO production was calculated according to the following equation.

$$FE_g(\%) = i_g/i_{tot} \times 100\% = (F \times x_g \times f_{CO2} \times Z_g)/i_{tot} \times 100\%$$

where $x_g$ is the molar flow of gas from the electrochemical cell (mol mol$^{-1}$), $f_{CO2}$ is the CO$_2$ flow rate (mol s$^{-1}$), $Z_g$ is the number of electron transferred for product formation, which is 2 for CO and H$_2$, the $i_{tot}$ is the total current in the electroreduction process. F is the Faradaic constant (96,485 C mol$^{-1}$).

Cyclic voltammetry (CV) curves in electrochemical double-layer capacitance (Cdl) determination were measured in a potential window nearly without the faradaic process at different scan rates of 10, 20, 30, 40, and 50 mV s$^{-1}$. The plot of current density at set potential against scan rate has a linear relationship and its slope is the Cdl.

### TEM sample preparation

The obtained Poly-(Au$_8$-DCP@Fe) was dispersed in 1 mL ethanol by ultrasound to form a suspension, which was then dropped on carbon film. The carbon film was dried under air within a few seconds and further used for TEM characterizations.

### In situ ATR-FTIR experiments

In-situ ATR-FTIR tests were implemented by Bruker INVENIO S FT-IR spectrophotometer equipped with a liquid N$_2$-cooled MCT detector. To enhance the in situ ATR-FTIR signal and electronic conduction, the ultra-thin gold foil was chemically deposited on the silicon. The working electrode for the in situ ATR-FTIR test was prepared by dropping the electrocatalyst on Au membrane.

### In situ CO-DRIFTS experiments

In situ CO-DRIFTS was performed on a Bruker INVENIO S FT-IR spectrophotometer equipped with an MCT narrow-band detector and a modified in situ reaction cell with a drier device. The detailed pretreatment and test conditions are given as follows. Firstly, the sample was carefully put into the support sheet of reaction cell, and a pure Ar stream (50 mL min$^{-1}$) was introduced for 20 min to remove the gaseous impurities in the cell. Subsequently, the sample was pre-reduced in a 5% H$_2$/Ar flow (50 mL min$^{-1}$) at 300 °C for 2 h, followed cooled 1.0 h to 25 °C in a pure Ar stream, and the background spectrum was collected in this process. Finally, the DRIFTS spectra was collected after introducing the 2% CO/N$_2$ (50 mL min$^{-1}$) to the cell for 1.0 h.

## Data availability

The data that support the findings of this study are available from the corresponding author upon reasonable request. All data needed to evaluate the conclusions in the paper are present in the paper and/or

the Supplementary Materials (including Supplementary Figs. 1–37). The X-ray crystallographic coordinates for structure reported in this article (see Supplementary Table 1) have been deposited at the Cambridge Crystallographic Data Centre (CCDC) under deposition number CCDC: 2290525 for $Au_8$. These data can be obtained free of charge from the Cambridge Crystallographic Data Centre via http://www.ccdc.cam.ac.uk/ data_request/cif.

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

## Acknowledgements

This work was supported by the National Natural Science Foundation of China (Nos. 22275168, 21825106, and 22175155).

## Author contributions

S.-Q.Z. proposed the research direction. S.W. guided the whole experiment. Y.-M.W. synthesized the catalysts, conducted the structure analysis and the catalytic studies. Q.-Y.W., C.-X.D., and F.-Q.Y. assisted in the completion of the whole experiment. L.-Y.W. and B.L. contributed to the revision process. All authors were active in writing this paper.

## Competing interests
The authors declare no competing interests.
