## [Peer Review File · Nature Communications]

REVIEWER COMMENTS

Reviewer #1 (Remarks to the Author):

In this work, the authors use single-atom (SA) sites to modify Au₈ clusters for electrochemical CO₂ reduction to CO. They found that the presence of SA sites (especially Fe) significantly improves the performance of Au₈ clusters. The designed catalyst shows a CO Faradaic efficiency of 90% and stable for about 6 hours in an H-cell configuration. While there are some interesting aspects about the material synthesis in this work, I believe it does not show notable advances in the field of electrochemical CO₂ reduction. The author chose Au₈ clusters as the active materials which show almost no CO₂ reduction activity (according to the result in the current manuscript). Thus, even though the “activity” of Au₈ clusters is significantly improved with the addition of SA sites, the overall performance of the designed catalysts is still far from those of the state-of-the-art Au based catalysts for CO₂ reduction. In addition, the stability test of 6 hours at a current density of 2.5 mA/cm², two orders of magnitude lower than industrially relevant current densities, is not attractive from an applied perspective. I have some major comments below:

1. The activity of Au clusters is often hindered by the presence of ligands on the surface. To maximize the activity of Au clusters, surface ligands are often removed before electrochemical reduction. In this work, the authors used Au₈ clusters with ligands covering their surface. Thus, it is expected that the activity of Au₈ clusters is very low. I would suggest the authors find a way to activate Au₈ clusters before combining them with SA sites.
2. The authors claim in the work that Au is the main active site in CO₂RR after introducing Fe. Previous published papers (Nano Lett. 2022, 22, 4, 1557–1565, Nano Energy 68 (2020) 104384,) have shown that tuning electrons localized around central atom in M-N single atom moiety (M: metal) facilitates efficient CO₂ activation and CO desorption. From this aspect, Fe sites could be activated and involved in CO₂ reduction when they are combined with Au clusters.
3. The authors highlighted the effect of Fe sites in tuning of the electronic state of Au species, which could promote the stabilization and reaction of intermediates in CO₂ conversion reaction. Therefore, the electronic state of Fe also needs to be studied carefully. According to the XPS spectra in Figure S10, it is hard to state that the binding energy of Fe2p in poly-(Au₈-DCP@Fe) was negatively shifted compared to poly(DCP-Fe) due to noise. In addition, the local coordination environment and chemical states comparison of Au and/or Fe species in both Poly-(Au₈-DCP) and Poly-(Au₈-DCP@Fe) samples need to be included to clarify the effect of Fe site's introduction in poly-(Au₈-DCP@Fe).
4. In order to compare with the poly-(Au₈-DCP@Fe) sample and make sure that Poly DCP@Fe includes only single atom sites, TEM analysis of Poly DCP@Fe needs to be added.
5. Figure 4b, c and d could not be found in the electrocatalytic CO₂ reduction discussion. Why was the Faradaic efficiency of CO of Poly Au₈ improved significantly after -0.57V vs RHE, contrasting to other samples' trend?
6. The Faradaic efficiency of CO decreases from 90% to 80% after only 6h. What causes this degradation?
7. The content of Fe and Au in all samples should be presented.

Reviewer #2 (Remarks to the Author):

This paper reports a study of CO₂ reduction to CO using a mixture of the Au₈ cluster and transition metal complexes prepared through a copolymerization process. Significantly enhanced electrocatalytic activity was observed for the mixture relative to the Au₈ cluster alone. The work may be of interest to Nature Comm, but it's not publishable in its current form. There are serious issues that need to be addressed before the work is suitable for publication.

1. The most important issue is, where does catalysis happen? On the Au₈ cluster or on the single atom sites of the transition metals? It is well known that the transition metal complexes are redox active and can be better catalytic sites than the inactive Au₈ cluster. The role of the Au₈ cluster needs to be clarified through control experiment. Its effect may be only to disperse the transition metal sites and play no catalytic roles. The cartoon in Fig. 5g is not a mechanism, it needs to be backed up by experimental evidence. Other substrates that should be considered include MOFs, or simply polymers without Au₈ at all or other gold clusters.
2. The title and abstract need to specify what CO₂ is reduced to (i.e. CO).
3. The abstract is not accurate by stating that Au₈ is modified by single atom sites. They are merely a mixture. Again, the role of Au₈ may be simply to disperse the transition metals.
4. There are more active recent Au nanocluster catalysts for CO₂RR to CO, which are much better understood and much more active (<https://doi.org/10.1021/jacs.2c00725>; <https://doi.org/10.1021/jacs.2c00789>). These results should be compared and discussed in the context of the current work.

This work reports a co-electropolymerization strategy to prepare nanocluster-single atom catalysts, and demonstrated a higher performance in electrocatalytic CO₂RR compared with the pristine metal nanoclusters and single atom. The detailed analysis revealed that the improved performance is derive from the effective regulation of the electronic structure of the nanoclusters by the single atom. The results are potentially interesting, and the work is quite thorough. Hence the manuscript is proposed for acceptance in Nature Communications after considering the following comments:

1. The work is novel in its current approach, and for the general reader, it would be useful to provide the mechanism of co-electropolymerization process.
2. The activity of the cluster is ascribed firstly to the structure of the cluster. However, no further discussion of the "active sites" in Au₈ nanocluster is included, which can be obtained from calculations, for example, from electrostatic potential surfaces.
3. There are contradicting statements in the manuscript. In Fig. 3b, the d-band center of the Au₈ crystal is closer to the Fermi level than that of the Poly-Au₈, but Poly-Au₈ does exhibit better catalytic activity than Au₈ crystal.
4. The calculation details are not provided, which make the reviewer skeptical about their computation results.
5. The data shown is not entirely showing the crystal data of the Au₈ nanocluster, such as the Au-Au bond lengths and bond angles.

Responses to Reviewers

Reviewer 1

In this work, the authors use single-atom (SA) sites to modify Au₈ clusters for electrochemical CO₂ reduction to CO. They found that the presence of SA sites (especially Fe) significantly improves the performance of Au₈ clusters. The designed catalyst shows a CO Faradaic efficiency of 90% and stable for about 6 hours in an H-cell configuration. While there are some interesting aspects about the material synthesis in this work, I believe it does not show notable advances in the field of electrochemical CO₂ reduction. The author chose Au₈ clusters as the active materials which show almost no CO₂ reduction activity (according to the result in the current manuscript). Thus, even though the “activity” of Au₈ clusters is significantly improved with the addition of SA sites, the overall performance of the designed catalysts is still far from those of the state-of-the-art Au based catalysts for CO₂ reduction. In addition, the stability test of 6 hours at a current density of 2.5 mA/cm², two orders of magnitude lower than industrially relevant current densities, is not attractive from an applied perspective. I have some major comments below:

We are grateful to **Reviewer 1** for valuable comments. As the CO₂ electrochemical reduction catalyst, the current density and stability of Poly-(Au₈-DCP@Fe) might be lower than the Au nanoparticles (Au NPs) based catalysts. While, Supplementary Fig. 33 and Supplementary Tab. S6 comprehensively compared CO₂RR performance with recently reported atomically precise Au nanoclusters (Au NCs) and bimetallic Au nanoclusters-based catalysts, Poly-(Au₈-DCP@Fe) had comparable performance in terms of CO FE and the applied potential.

In our work, we proposed a novel eletropolymerization strategy to synthesis single atoms and metal clusters hybrid catalyst. The atomically controllable features of metal NCs and single atoms (SA) in this strategy enable fine tuning of the overall activity of catalysts. Moreover, the full predispersed metal NCs and SA monomers in hybrid catalyst provide an ideal platform to understand the critical roles of NCs and SAs in catalytic processes.

1) The activity of Au clusters is often hindered by the presence of ligands on the surface. To maximize the activity of Au clusters, surface ligands are often removed before electrochemical reduction. In this work, the authors used Au₈ clusters with ligands covering their surface. Thus, it is expected that the activity of Au₈ clusters is very low. I would suggest the authors find a way to activate Au₈ clusters before combining them with SA sites.

Response: Thanks for the constructive suggestions. We appreciate your insights regarding the potential negative impact of surface ligands on nanocluster catalysis. While it is commonly employed to remove surface protecting ligands by calcination, this can alter the nanocluster's core geometry, potentially leading to particle coalescence and hindering the correlation between structures and properties. Moreover, ligand removal may introduce complex metal–support interactions.

In response to your concerns, we want to emphasize that our catalyst preparation strategy differs from the conventional approach. Instead of removing surface ligands, our process involves

electrochemical polymerization of carbazoles on the Au₈ cluster, facilitating integration with Fe single atom sites. This unique method ensures the preservation of the nanocluster's structure while achieving the desired functionality.

Our findings reveal that the surface carbazole ligands in the polymerized hybrid network positively impact CO₂ reduction catalysis. Unlike discrete Au₈ clusters, the electropolymerized biscarbazole network enhances electron-transfer to metal catalytic sites, extending the active surface area far beyond the electrode. Electrochemical double-layer capacitance measurements confirm that Poly-Au₈ exhibits a significantly higher electrochemical surface area compared to pristine Au₈cpz crystal, underscoring its superior performance in providing active sites for CO₂ reduction.

Accordingly, we have incorporated the above discussions into the revised supporting information (Page 23, the last paragraph).

- 2) The authors claim in the work that Au is the main active site in CO₂RR after introducing Fe. Previous published papers (Nano Lett. 2022, 22, 4, 1557–1565, Nano Energy 68 (2020) 104384,) have shown that tuning electrons localized around central atom in M-N single atom moiety (M: metal) facilitates efficient CO₂ activation and CO desorption. From this aspect, Fe sites could be activated and involved in CO₂ reduction when they are combined with Au clusters.

Response: Thank you for providing insightful query and these important references. We concur with your observation regarding the role of localized electrons in M-N-C catalysts for enhancing CO₂RR selectivity and activity. While M-N single atom catalysts in previous studies demonstrated inherent catalytic activity, our system, particularly the single-atom catalysts (Poly-DCP@Fe) and DCP@Fe control samples, exhibited minimal activity in electrocatalytic CO₂ to CO reduction. In addition, theoretical calculations reinforce the prominence of Au₈ NCs over Fe SA sites in driving CO₂RR. Higher energy barrier for electrochemical CO₂RR and lower energy barrier for HER with Fe SA sites compared to Au₈ NCs sites in Poly-(Au₈-DCP@Fe) indicate that the catalytic efficiency of CO₂RR primarily stems from Au₈ NC sites. Consistent trends observed in the free energy calculations for different single atom (Poly-(Au₈-DCP@M), M = Co, Ni, Cu, Zn) further support the superior activity of Au₈ clusters in CO₂RR (Figure R1).

Considering the inactivity of Fe single-atom sites in our system, literature consultation leads us to hypothesize that the high N-coordinated configuration, exemplified by Fe-N₆ coordination, might contribute to this observed inactivity. This configuration appears less conducive to CO₂-to-CO conversion due to its diminished charge transfer efficiency and the hindered adsorption and activation of CO₂ to form the *COOH intermediate, as supported by references (Ref. *Nat. Commun.* **14**, 5245 (2023); *Green Chem.* **22**, 7529-7536 (2020)).

Figure R1. The free energy diagram of the (a-d) CO₂RR and (e-h) HER on the M sites and Au₈ sites of Poly-(Au₈-DCP@M), M = Co, Ni, Cu, Zn.

Accordingly, to enhance clarity and accessibility, we have thoughtfully integrated the derived results and insightful discussions into the revised manuscript (Page 17, the first paragraph) and the Supporting Information (Figure S35).

3) The authors highlighted the effect of Fe sites in tuning of the electronic state of Au species, which could promote the stabilization and reaction of intermediates in CO₂ conversion reaction. Therefore, the electronic state of Fe also needs to be studied carefully. According to the XPS spectra in Figure S10, it is hard to state that the binding energy of Fe2p in poly-(Au₈-DCP@Fe) was negatively shifted compared to poly-(DCP-Fe) due to noise. In addition, the local coordination environment and chemical states comparison of Au and/or Fe species in both Poly-(Au₈-DCP) and Poly-(Au₈-DCP@Fe) samples need to be included to clarify the effect of Fe site's introduction in poly-(Au₈-DCP@Fe).

Response: We express our gratitude to **Reviewer 1** for the valuable comments, which have substantially enhanced the quality of our manuscript. In response, we conducted a meticulous reevaluation of XPS for Poly-DCP@Fe and Poly-(Au₈-DCP@Fe). Despite the inherent surface probing nature of XPS and the challenge posed by the encapsulation of Fe single atom sites within the polymer network post-electrochemical polymerization, the obtained data (Figure R2) reveals discernible trends. Specifically, the observed negative shift in the binding energies of Fe 2p in Poly-(Au₈-DCP@Fe) compared to Poly-DCP@Fe signifies a reduction in the valence state of Fe SAs. In addition, this work further employs CO-DRIFTS probe analysis to illustrate the alterations in the electronic state of Poly-(Au₈-DCP@Fe) in comparison with Poly-(DCP@Fe). To elucidate the influence of Fe sites on the electronic state of Au species, we delved into Au L₃-edge X-ray absorption near-edge structure (XANES) results for Poly-Au₈, Poly-(Au₈-DCP) and Poly-(Au₈-DCP@Fe). As depicted in Figure R3, the proximity of the white lines in Poly-Au₈ and Poly-(Au₈-DCP@Fe), corroborated by XPS spectra, suggests a close resemblance. The shift in the near-edge of Au L₃-edge spectra for Poly-(Au₈-DCP@Fe) towards higher energy indicates an elevated oxidation state of Au atoms after the introduction of Fe single atoms, aligning with the finding from XPS.

Figure R2. XPS spectra of Fe 2p for (a) Poly-(Au₈-DCP@Fe) and (b) Poly-DCP@Fe.

Figure R3. The Au L₃-edge XANES of Au foil, Poly-Au₈, Poly-(Au₈-DCP) and Poly-(Au₈-DCP@Fe).

We really appreciate **Reviewer 1** for the invaluable suggestions, which have significantly strengthened our paper. Accordingly, we have integrated the aforementioned insights into the revised manuscript (Page 8, the first paragraph) and Supporting Information (Figures S9 and S13).

- 4) In order to compare with the poly-(Au₈-DCP@Fe) sample and make sure that Poly-DCP@Fe includes only single atom sites, TEM analysis of Poly DCP@Fe needs to be added.

Response: Thanks for the good suggestion. The requested TEM analysis and HAADF-STEM image of Poly-DCP@Fe are now provided in the supplementary Figures S7-S8 and are also showcased as Figures R4-5 below. The TEM image affirms the absence of metal nanoparticles. The HAADF-STEM image, as depicted in Figure R5, distinctly reveals atomically dispersed Fe species through atom-scale bright spots, confirming their isolated single metal atom configuration.

Figure R4. (a) TEM image and (b) Elemental mapping of Poly-DCP@Fe.

Figure R5. (a) Atomic-resolution HAADF-STEM image of Poly-DCP@Fe revealing the atomically dispersed Fe SA. (b) Elemental mapping of overlapped images and C (red), Fe (cyan) and N (blue).

Accordingly, we have added the above results and discussion into the revised manuscript (Page 7, the second paragraph) and Supporting Information (Figures S7-8).

- 5) Figure 4b, c and d could not be found in the electrocatalytic CO₂ reduction discussion. Why was the Faradaic efficiency of CO of Poly-Au₈ improved significantly after -0.57 V vs RHE, contrasting to other samples' trend?

Response: We thank the **Reviewer** for pointing out these typos, and we have rectified the oversight by revising 'Figure 2b, c and d' to 'Figure 4b, c and d' in the electrocatalytic CO₂ reduction discussion.

Addressing the concern about the CO Faradaic efficiency of Poly-Au₈ improving significantly after -0.57 V vs RHE, repeated experiments consistently demonstrate a similar trend. *Quasi-in situ* XPS of Poly-Au₈ revealing changes during the CO₂RR process at -0.57 V, -0.67 V, and -0.77 V (Figure R6). The broadening of P 2p peak after -0.57 V vs RHE suggests the formation of defects due to partial removal of bisphosphine ligands, exposing catalytic sites and enhancing CO₂RR efficiency. Additionally, the shift in P 2p to lower binding energies indicates an increased electron cloud density of ligands, which supports stripping off the strong electron-donated bidentate bisphosphine ligands. Additionally, Figure R7a illustrates that,

compared to Poly-(Au₈-DCP@Fe), Poly-Au₈ displays higher overpotentials and smaller current densities in CO₂-saturated electrolyte, and shows a cathodic current response just below 1 mA cm⁻² at -0.47 V and -0.57 V vs. RHE under CO₂ (Figure R7b, c). Under such a small current density, hydrogen evolution reaction (HER) tends to dominate (Ref. *Nat. Catal.* **5**, 268-276 (2022); *J. CO₂ Util.* **30**, 168-182 (2019)), hindering CO₂ to CO conversion and resulting in a low Faraday efficiency of CO for Poly-Au₈. Notably, at -0.67 V, a sharp increase in current density (3 mA cm⁻²) leads to a significant increase in CO Faraday efficiency for Poly-Au₈.

Regarding Poly-(Au₈-DCP) and Poly-(Au₈-DCP@Fe), the introduction of DCP forming an electron conducting network enhances the catalytic activity of Poly-Au₈ by facilitating electron transfer and optimizing catalytic site utilization. Therefore, compared to Poly-Au₈, both Poly-(Au₈-DCP) and Poly-(Au₈-DCP@Fe) exhibit higher current density and CO Faraday efficiency at lower potentials.

Figure R6. (a) Schematic illustration of Poly-Au₈ partial bidentate pisphosphine ligands removal at -0.67 V and -0.77 V. (b) The *Quasi-in situ* XPS spectra of P 2p for Poly-Au₈ at -0.57 V, -0.67 V and -0.77 V (vs. RHE).

Figure R7. (a) LSV curves for Poly-Au₈ and Poly-(Au₈-DCP@Fe) in a CO₂-saturated KHCO₃ solution at the potential range of 0 to -1.17 V (vs RHE, scan rate of 50 mV/s). (b, c) Chronoamperometric responses of (b) Poly-(Au₈-DCP@Fe) and (c) Poly-Au₈ at different potentials (-0.47 to -0.87 V, vs. RHE).

Accordingly, we have added the above discussion into the revised manuscript (Page 13, the College of Chemistry, Zhengzhou University, Zhengzhou 450001, China

first paragraph) and Supporting Information (Figure S22-23).

- 6) The Faradaic efficiency of CO decreases from 90% to 80% after only 6h. What causes this degradation?

Response: In response to this insightful suggestion from the reviewer, a thorough examination of the HAADF-STEM image of Poly-(Au₈-DCP@Fe) after the stability test unveiled a noteworthy observation. The size of certain Au nanoclusters exhibited a discernible increase, growing from approximately 0.93 nm to around 1.27 nm. Considering the well-established correlation between nanocluster size and catalytic performance, we posit that the observed decline in catalyst efficiency can be attributed to the enlargement of these nanoclusters during the extended stability test.

Figure R8. HAADF-STEM image of Poly-(Au₈-DCP@Fe) after stability test.

Accordingly, we have added the above results and discussion into the revised manuscript (Page 14, the first paragraph) and Supporting Information (Figure S30).

- 7) The content of Fe and Au in all samples should be presented.

Response: Following this suggestion, the content of Fe and Au in all samples have been determined by ICP-OES analysis and provided in supplementary Table S4, and are also shown as Table R1 below.

Table R1. The weight percentage of Au or Fe element in Au₈ crystal, Poly-Au₈, Poly-(Au₈-DCP), Poly-(Au₈-DCP@Fe), Poly-DCP@Fe and DCP@Fe.

	Au%	Fe%
Au ₈ crystal	42.58	–
Poly-Au ₈	45.24	–
Poly-(Au ₈ -DCP)	8.92	–
Poly-(Au ₈ -DCP@Fe)	5.99	4.00
Poly-DCP@Fe	–	3.56
DCP@Fe	–	4.68

We thank **Reviewer 1** for the valuable suggestions for improving our manuscript.

Reviewer 2

This paper reports a study of CO₂ reduction to CO using a mixture of the Au₈ cluster and transition metal complexes prepared through a copolymerization process. Significantly enhanced electrocatalytic activity was observed for the mixture relative to the Au₈ cluster alone. The work may be of interest to Nature Comm, but it's not publishable in its current form. There are serious issues that need to be addressed before the work is suitable for publication.

- 1) The most important issue is, where does catalysis happen? On the Au₈ cluster or on the single atom sites of the transition metals? It is well known that the transition metal complexes are redox active and can be better catalytic sites than the inactive Au₈ cluster. The role of the Au₈ cluster needs to be clarified through control experiment. Its effect may be only to disperse the transition metal sites and play no catalytic roles. The cartoon in Fig. 5g is not a mechanism, it needs to be backed up by experimental evidence. Other substrates that should be considered include MOFs, or simply polymers without Au₈ at all or other gold clusters.

Response: Thank you for your insightful comments. We fully understand **Reviewer 2's** concerns regarding the catalytic activity on Au clusters versus Fe single-atoms. In addressing this concern, we have conducted a series of controlled experiments and theoretical calculations, clearly demonstrating that metal nanoclusters, particularly Au₈ clusters, serve as more likely catalytic active sites than Fe single-atoms.

To further substantiate our conclusion, we introduced 4,4'-Di(9H-carbazol-9-yl)-1,1'-biphenyl (BCP) containing two carbazole groups, a comonomer resembling Au₈ clusters, for co-electropolymerization with DCP@Fe (Figure R9a). The resulting catalyst (Poly-(DCP@Fe-BCP)), as confirmed by TEM imaging, exhibited no noticeable particles or clusters (Figure R10). However, its electrocatalytic CO₂RR performance, presented in Figure R9b, revealed predominant H₂ production across a wide potential range from -0.47 to -0.87 V (vs. RHE) with H₂ FE near 100%, indicating that dispersing Fe single-atoms alone does not activate the catalyst for CO₂-to-CO conversion. Moreover, to emphasize the pivotal role of Au₈ clusters as likely active sites, we prepared Poly-(Au₈-BCP) by co-electropolymerization of Au₈ clusters with BCP, resulting in uniformly dispersed Au NCs (Figure R11). The electrocatalytic CO₂RR performance of Poly-(Au₈-BCP), showcased in Figure R9c, exhibited significantly higher activity (CO FE_{maximum} = 78.32% at -0.57 V) compared to Poly-(DCP@Fe-BCP). This affirms that sufficiently dispersed Au₈ clusters are more likely to act as catalytic active sites, triggering the catalyst's activity for CO₂-to-CO conversion.

In parallel, we evaluated the catalytic performance of the control material DCP@Fe, as presented in Figure R12. Similar to Poly-DCP@Fe, DCP@Fe primarily produced H₂ with H₂ FE near 100% over a broad potential range, reinforcing the conclusion that Fe SAs in these catalytic systems is inactive in the electrocatalytic CO₂ to CO reaction.

Figure R9. (a) Schematic illustration of Poly-(DCP@Fe-BCP) and Poly-(Au₈-BCP) fabricated by co-electropolymerization strategy. Faradaic efficiencies of (b) Poly-(DCP@Fe-BCP) and (c) Poly-(Au₈-BCP) at different applied potentials in CO₂-saturated 0.5 M KHCO₃ aqueous solution

Figure R10. (a) TEM image and (b) Elemental mapping of Poly-(DCP@Fe-BCP).

Figure R11. (a) TEM image and (b) Elemental mapping of Poly-(Au₈-BCP).

Figure R12. (a) LSV curves for DCP@Fe in a CO₂-saturated and Ar-saturated KHCO₃ solution at the potential range of 0 to -1.17 V (vs RHE, scan rate of 50 mV/s). (b) Faradaic efficiencies of DCP@Fe at different applied potentials in CO₂-saturated 0.5 M KHCO₃ aqueous solution.

As suggested by **Reviewer 2**, we have included the above results and discussions in the revised manuscript (Page 13, the first paragraph) and Supporting Information (Figure S24-27).

2) The title and abstract need to specify what CO₂ is reduced to (i.e. CO).

Response: Thank you for the good comment. Based on your suggestion, the title of this manuscript has been revised to ‘Single-Atom Tailored Atomically-Precise Nanoclusters for Enhanced Electrochemical Reduction of CO₂-to-CO Activity’. In the abstract, we also specify the CO product of CO₂ reduction.

3) The abstract is not accurate by stating that Au₈ is modified by single atom sites. They are merely a mixture. Again, the role of Au₈ may be simply to disperse the transition metals.

Response: Thanks for the valuable comments. As discussed in question 1, we have clarified that dispersing Fe single-atoms does not enhance catalyst selectivity or activity for CO₂-to-CO conversion; metal nanoclusters are identified as more likely catalytic active sites than single-atoms. Therefore, we believe that the statement of "Au₈ NCs modified by single-atom sites" is more reasonable.

4) There are more active recent Au nanocluster catalysts for CO₂RR to CO, which are much better understood and much more active (<https://doi.org/10.1021/jacs.2c00725>; <https://doi.org/10.1021/jacs.2c00789>). These results should be compared and discussed in the context of the current work.

Response: We appreciate the **Reviewer 2** for providing these important literatures. In response to the suggestion, we have enriched the revised version with additional discussions and pertinent references (Page 14, the last paragraph, Refs. 51-52). The expanded discussions are included as follows: “In comparison with recently reported Au NCs based catalysts, such as Au₂₂H₃ (CO FE_{maximum} = 92.7 % at -0.60 V)⁵¹ and Au₂₄H₃ (CO FE_{maximum} > 90 % at 10 mA/cm⁻²)⁵², Poly-(Au₈-DCP@Fe) (CO FE_{maximum} = 90.89 % at -0.57 V) demonstrated comparable CO₂RR performance in terms of CO FE and the applied potential. A comprehensive comparison, presented in Supplementary Fig. 33 and Supplementary Tab. S6, showcased the CO₂RR performance of Poly-(Au₈-DCP@Fe) against reported Au NCs and bimetallic Au NCs based catalysts, underscoring the exceptional activity of Poly-(Au₈-DCP@Fe).”

We sincerely appreciate **Reviewer 2** for the valuable suggestions for improving our manuscript.

Reviewer 3

This work reports a co-electropolymerization strategy to prepare nanocluster-single atom catalysts, and demonstrated a higher performance in electrocatalytic CO₂RR compared with the pristine metal nanoclusters and single atom. The detailed analysis revealed that the improved performance is derive from the effective regulation of the electronic structure of the nanoclusters by the single atom. The results are potentially interesting, and the work is quite thorough. Hence the manuscript is proposed for acceptance in Nature Communications after considering the following comments.

- 1) The work is novel in its current approach, and for the general reader, it would be useful to provide the mechanism of co-electropolymerization process.

Response: We appreciate the reviewer for the encouragement of our work and the valuable comment. Following this suggestion, we have added the mechanism of carbazole monomer oxidation, crosslinking, and reduction during the CV scans, and included them in the revised Figure S1.

Figure R13. Mechanism of monomer oxidation, crosslinking, and reduction during the CV scans.

- 2) The activity of the cluster is ascribed firstly to the structure of the cluster. However, no further discussion of the "active sites" in Au₈ nanocluster is included, which can be obtained from calculations, for example, from electrostatic potential surfaces.

Response: Thank you very much for the constructive comments. Based on your suggestion, we have employed the calculations of the surface electrostatic potential to discuss the catalytically active sites in Au₈ nanocluster. As showed in Figure R14, negative ESP regions surrounding the organic moieties of surface ligands drive robust interactions with CO₂ molecules, elevating CO₂ concentration near active sites for enhanced CO₂ conversion. Furthermore, DFT calculations pinpoint the *COOH intermediate adsorption site on Poly-(Au₈-DCP@Fe) to be the No. 9 Au atom, indicating that the catalytic site for CO₂ reduction predominantly resides at this specific location.

Figure R14. (a) Electrostatic potential, V_{el} , mapped onto the charge density isosurface for Poly-(Au₈-DCP@Fe) and (b) Calculated *COOH adsorption site on Poly-(Au₈-DCP@Fe) structure.

Accordingly, we have included the above data and discussion in the revised manuscript (Page 17, the first paragraph) and Supporting Information (Figure S37).

- 3) There are contradicting statements in the manuscript. In Fig. 3b, the d-band center of the Au₈ crystal is closer to the Fermi level than that of the Poly-Au₈, but Poly-Au₈ does exhibit better catalytic activity than Au₈ crystal.

Response: Thanks for your valuable comment. In our DFT calculations, we utilized a simplified structural unit of Poly-Au₈, consisting of an Au₈ core with biscarbazole as ligands. The close resemblance between the Poly-Au₈ and Au₈ crystal models results in minimal disparity in d-band center calculations. However, it's essential to note that the experimental Poly-Au₈ catalyst is a complex polymer with a high molecular weight. Thus, slight variations between experimental results and DFT calculations are expected, given the inherent distinctions between the modeled structure and the actual material used for measurements.

- 4) The calculation details are not provided, which make the reviewer skeptical about their computation results.

Response: Thank you for your valuable comment. As the Reviewer 3's suggested, the calculation details have been incorporated in the revised supporting information. All the calculations were performed within the framework of the density functional theory (DFT) as implemented in the Vienna Ab initio Software Package (VASP 5.4.4) code within the Perdew–Burke–Ernzerhof (PBE) generalized gradient approximation and the projected augmented wave (PAW) method. The cutoff energy for the plane-wave basis set was set to 450 eV. The Brillouin zone of the surface unit cell was sampled by Monkhorst–Pack (MP) grids, with k-point mesh density of $2\pi \times 0.04 \text{ \AA}^{-1}$ for structures optimizations. The convergence criterion for the electronic

self-consistent iteration and force was set to 10^{-5} eV and 0.01 eV/Å, respectively. The vacuum layer of 15 Å was introduced to avoid interactions between periodic images.

The free energies of adsorbates at temperature T were estimated according to the harmonic approximation, and the entropy is evaluated using the following equation:

$$S(T) = K_B + \sum_i^{\text{harm DOF}} \left(\frac{\varepsilon_i}{K_B T (e^{\frac{\varepsilon_i}{K_B T}} - 1)} - \ln \left(1 - e^{-\frac{\varepsilon_i}{K_B T}} \right) \right)$$

where K_B is Boltzmann's constant and DOF is the number of harmonic energies (ε_i) used in the summation denoted as the degree of freedom, which is generally $3N$, where N is the number of atoms in the adsorbates. Meanwhile, the free energies of gas phase species are corrected as:

$$G_g(T) = E_{elec} + E_{ZPE} + \int C_p dT - TS(T)$$

where C_p is the gas phase heat capacity as a function of temperature derived from Shomate equations and the corresponding parameters in the equations were obtained from NIST.

Accordingly, we have included the calculations details in the revised Supporting Information (Page 13).

5) The data shown is not entirely showing the crystal data of the Au_8 nanocluster, such as the Au-Au bond lengths and bond angles.

Response: Thanks for your good comment. The requested crystal data of Au_8 nanocluster are now provided in the supplementary Table S2 and Table S3, and are also shown as Table R2 and R3 below.

Table R2. Bond lengths for Au_8cpz .

	Au-Au bond	Bond length (Å)
	Au1-Au1'	2.6178(2)
	Au1-Au2	2.84212(18)
	Au1-Au2'	2.85219(18)
	Au1-Au3'	2.90083(18)
	Au1-Au3	2.81354(19)
	Au2-Au1'	2.85216(18)
	Au2-Au3'	2.64965(19)
	Au2-Au4	3.0677(2)
	Au3-Au1'	2.90082(18)
	Au3-Au2'	2.64966(19)
	Au3-Au4'	3.1028(2)
	Au4-Au3'	3.1027(2)

Table R3. Bond angles for Au_8cpz .

	Atom-Atom-Atom	Angle/°
	Au1'-Au1-Au2	62.827(6)
	Au1'-Au1-Au2'	62.437(5)
	Au1'-Au1-Au3'	61.048(6)
	Au1'-Au1-Au3	64.447(6)
	Au2-Au1-Au2'	125.264(5)
	Au2-Au1-Au3'	54.940(5)
	Au2'-Au1-Au3'	97.363(5)
	Au3-Au1-Au2'	55.761(5)
	Au3-Au1-Au2	99.632(6)
	Au3-Au1-Au3'	125.495(5)
	Au1-Au2-Au1'	54.737(5)
	Au1'-Au2-Au4	123.612(6)
	Au1-Au2-Au4	113.904(6)
	Au3'-Au2-Au1	63.656(5)
	Au3'-Au2-Au1'	61.380(5)
	Au3'-Au2-Au4	65.254(5)
	Au1-Au3-Au1'	54.503(5)
	Au1-Au3-Au4'	123.712(6)
	Au1'-Au3-Au4'	111.214(6)
	Au2'-Au3-Au1	62.858(5)
Au2'-Au3-Au1'	61.405(5)	
Au2'-Au3-Au4'	63.888(5)	
Au2-Au4-Au3'	50.856(4)	

We thank **Reviewer 3** for the valuable suggestions for improving our manuscript.

REVIEWERS' COMMENTS

Reviewer #1 (Remarks to the Author):

The authors have addressed my concerns.

Reviewer #2 (Remarks to the Author):

The authors have done a good job in carrying out further experiments to address the concerns raised by the reviewers. I have no further comments and would be happy to recommend acceptance.

Reviewer #3 (Remarks to the Author):

This manuscript has been improved and might be accepted.